# MAIT cells are functionally impaired in a Mauritian cynomolgus macaque model of SIV and Mtb co-infection

Amy L. Ellis[1], Alexis J. Balgeman[1], Erica C. Larson[2], Mark A. Rodgers[2], Cassaundra Ameel[2], Tonilynn Baranowski[2], Nadean Kannal[1], Pauline Maiello[2], Jennifer A. Juno[3], Charles A. Scanga[2], Shelby L. O'Connor[1]*

**1** Department of Pathology and Laboratory Medicine, University of Wisconsin-Madison, Madison, Wisconsin, United States of America, **2** Department of Microbiology and Molecular Genetics, and Center for Vaccine Research, University of Pittsburgh, Pittsburgh, Pennsylvania, United States of America, **3** Department of Microbiology and Immunology, University of Melbourne, Melbourne, Australia

* slfeinberg@wisc.edu

**Data Availability Statement:** All relevant data are within the manuscript and its Supporting Information files.

## Abstract

Mucosal-associated invariant T (MAIT) cells can recognize and respond to some bacterially infected cells. Several *in vitro* and *in vivo* models of *Mycobacterium tuberculosis* (Mtb) infection suggest that MAIT cells can contribute to control of Mtb, but these studies are often cross-sectional and use peripheral blood cells. Whether MAIT cells are recruited to Mtb-affected granulomas and lymph nodes (LNs) during early Mtb infection and what purpose they might serve there is less well understood. Furthermore, whether HIV/SIV infection impairs MAIT cell frequency or function at the sites of Mtb replication has not been determined. Using Mauritian cynomolgus macaques (MCM), we phenotyped MAIT cells in the peripheral blood and bronchoalveolar lavage (BAL) before and during infection with SIV-mac239. To test the hypothesis that SIV co-infection impairs MAIT cell frequency and function within granulomas, SIV+ and -naïve MCM were infected with a low dose of Mtb Erdman, and necropsied at 6 weeks post Mtb-challenge. MAIT cell frequency and function were examined within the peripheral blood, BAL, and Mtb-affected lymph nodes (LN) and granulomas. MAIT cells did not express markers indicative of T cell activation in response to Mtb *in vivo* within granulomas in animals infected with Mtb alone. SIV and Mtb co-infection led to increased expression of the activation/exhaustion markers PD-1 and TIGIT, and decreased ability to secrete TNFα when compared to SIV-naïve MCM. Our study provides evidence that SIV infection does not prohibit the recruitment of MAIT cells to sites of Mtb infection, but does functionally impair those MAIT cells. Their impaired function could have impacts, either direct or indirect, on the long-term containment of TB disease.

## Author summary

MAIT cells are a population of immune cells that can directly detect and destroy some bacterially infected cells. Evidence suggests that MAIT cells may play a role in control of

**Funding:** Shelby L. O'Connor (SLO) received the following funding from the National Institutes of Health (NIH): R21 AI127127 (URL: www.nih.gov). Charles A. Scanga (CAS) received the following funding from the National Institutes of Health (NIH): NIH RO1 AI-111815 (URL: www.nih.gov). The sponsors/funders did not play any role in study design, data collection and analysis, decision to publish, or preparation of the manuscript.

**Competing interests:** The authors have declared that no competing interests exist.

*Mycobacterium tuberculosis* (Mtb) infection, but few studies have examined MAIT cell activity within granulomas, which are the sites of Mtb replication. In addition, chronic HIV infection has been shown to impair the frequency and function of MAIT cells in humans, but these studies focus on peripheral blood and not the sites of Mtb infection. Here, we used a macaque model of SIV and Mtb co-infection to determine whether SIV, as a model for HIV, could dysregulate MAIT cells in tissues where Mtb replication is occurring. SIV co-infection did not affect the absolute numbers of MAIT cells present within granulomas but did impair the ability of the MAIT cells to respond to mycobacteria both *in vivo* and *in vitro*. Overall, our study provides evidence that SIV infection alters MAIT cells phenotypically, and impairs MAIT cell function. What effect this might have on antimycobacterial immunity is an avenue for future exploration.

## Introduction

*Mycobacterium tuberculosis* (Mtb) is the causative agent of tuberculosis (TB), and 10 million new cases of TB occurred in 2018 alone [1]. Ninety percent of healthy humans are able to immunologically control Mtb infection; however, TB remains a major global health concern. One factor that can complicate the outcome of Mtb infection is co-infection with human immunodeficiency virus (HIV). HIV+ individuals are 20 times more likely to develop active TB disease and Mtb infection is the most common cause of death in HIV+ individuals [2, 3].

We do not understand fully the extent of immune responses that are dysregulated by HIV and contribute to the weakened control of Mtb, particularly within granulomas and Mtb-affected lymph nodes. [4]. Depletion of CD4 T cells is a hallmark of HIV infection. These cells are also important for Mtb control, but the mechanism by which they control Mtb infection is still not fully understood [5, 6]. Macaque studies have also shown that animals can maintain control of latent Mtb infection, even when CD4 T cells are depleted [7, 8]. Further, HIV+ individuals treated with antiretroviral therapy with recovered peripheral CD4 T cell counts are still at higher risk for TB, when compared to those who are HIV-naïve [9]. Together, these data indicate that there must be non-CD4 T cell immune responses that are important for Mtb control.

The contribution of mucosal associated invariant T (MAIT) cells to Mtb control is not well understood. MAIT cells are a specialized population of T lymphocytic cells that recognize riboflavin metabolites produced by some intracellular bacteria and presented by MR1 molecules [10–13]. In humans and macaques, MAIT cells are highly abundant in the lungs and bronchoalveolar lavage (BAL) fluid, and express activation markers, such as CD69, in the blood following Mtb infection [14, 15]. In humans, MAIT cell frequencies are reduced in the blood during active TB [13, 16–18]. Several *in vitro* studies have demonstrated the ability of MAIT cells to respond to mycobacterial antigens [11, 14, 19, 20]. Furthermore, MR1 deficient mice develop exacerbated TB [13, 21]. However, a recent study performed in Mtb-infected rhesus macaques indicated that MAIT cells within the tissues or airways may not respond to Mtb infection [22]. Indeed, little evidence exists to demonstrate that MAIT cells respond to Mtb within the granulomas and affected LN.

The impact of HIV or SIV infection on MAIT cell frequency and function remains under debate. Several cross-sectional human studies indicate that MAIT cells are depleted during HIV infection, but the mechanism by which this occurs is not well understood. The depletion has been suggested to occur from immune over-activation and subsequent apoptosis, as well as possibly via downmodulation of surface receptors [17, 18, 23–26]. Similar to studies of HIV

+ humans, the frequency of MAIT cells was also lower in one cross-sectional study of SIV+ rhesus macaques, when compared to SIV-naïve controls [27]. These cross-sectional studies of HIV/SIV+ individuals were challenged by recent longitudinal studies of HIV+ humans and SIV+ pig-tailed macaques that found MAIT cells were not depleted during the first year of infection [28, 29]. Notably, most of these studies only measured MAIT cells in the blood. A recent report examining MAIT cells within the lungs of HIV+ and HIV-naïve patients with active TB found that MAIT cell frequency was actually higher in the lungs of the HIV+ patients compared to the HIV-naïve patients [30]. Overall, few studies to date have focused on how HIV/SIV infection affects MAIT cell frequency and/or function within granulomas of Mtb-infected individuals.

We previously established a macaque model of SIV and Mtb co-infection to study how pre-existing SIV infection impairs the host immune response to Mtb. For these studies we used Mauritian cynomolgus macaques (MCM), which have simplified major histocompatibility complex (MHC) genetics [31]. Because of the simplified genetics, the adaptive immune response is easier to characterize as many animals will have similar CD4 and CD8 T cell responses to Mtb infection. We previously used MCM to generate tetramer reagents and characterize Mtb-specific adaptive immune cell responses [32]. We found that SIV/Mtb co-infected animals exhibited a more rapid increase in the number of granulomas between 4 and 8 weeks post Mtb challenge, when compared to those infected with Mtb alone [33]. This observation suggested that SIV infection impaired the ability of the co-infected MCM to contain Mtb infection some time between 4 and 8 weeks post-Mtb challenge.

In this study, we hypothesized that SIV infection would impair MAIT cell frequency and function in Mtb-affected granulomas and LNs between 4 and 8 weeks after infection, when compared to MAIT cells present in SIV-naïve animals. To test this hypothesis MCM were infected with SIVmac239 for 6 months, followed by co-infection with a low dose of Mtb for 6 weeks. We included an SIV-naïve control group that was infected with only Mtb. There were no striking clinical differences in TB progression between the two groups at necropsy just 6 weeks post Mtb. We also included another control group of animals that were infected with SIVmac239 and then euthanized at 6 months after infection, at the time when they would otherwise be challenged with Mtb. This control group was important to characterize MAIT cells in the tissues of SIV+ animals, without being confounded by Mtb co-infection. Using these groups of animals, we could determine whether SIV infection disrupted the frequency, phenotype, and/or function of MAIT cells in the blood and tissues, thus weakening their ability to contain the initial Mtb infection.

## Results

### Characterization of MAIT cells in SIV-naïve MCM

We first defined the phenotype of MAIT cells in SIV-naïve MCMs. Frozen peripheral blood mononuclear cells (PBMC) from 17 SIV-naïve MCM were stained with the Mamu-MR1-5OP-RU tetramer. The MR1 tetramer-positive cells in the blood, henceforth referred to as MAIT cells, were then further characterized. The gating schematic for MAIT cells is shown in Fig 1A. MAIT cells comprised an average of 1.36% of the CD3+ T cells across the MCM tested. All of the circulating MAIT cells were CD8+ in all animals examined, and no CD4+, CD4-CD8-, or CD8+CD4+ MAIT cells were observed (Fig 1A). This was surprising given the fact that in humans and mice, MAIT cells are more diverse in their CD4 and CD8 expression [34–36]. The majority (>90%) of MAIT cells expressed TCRVα7.2 (Fig 1A; right panel) [35]. Given the predominant CD8 expression we observed, the rest of our analysis was performed on CD8+MR1tet+ cells (Fig 1B). MAIT cells comprised a wide range (0.6%-17%) of the bulk

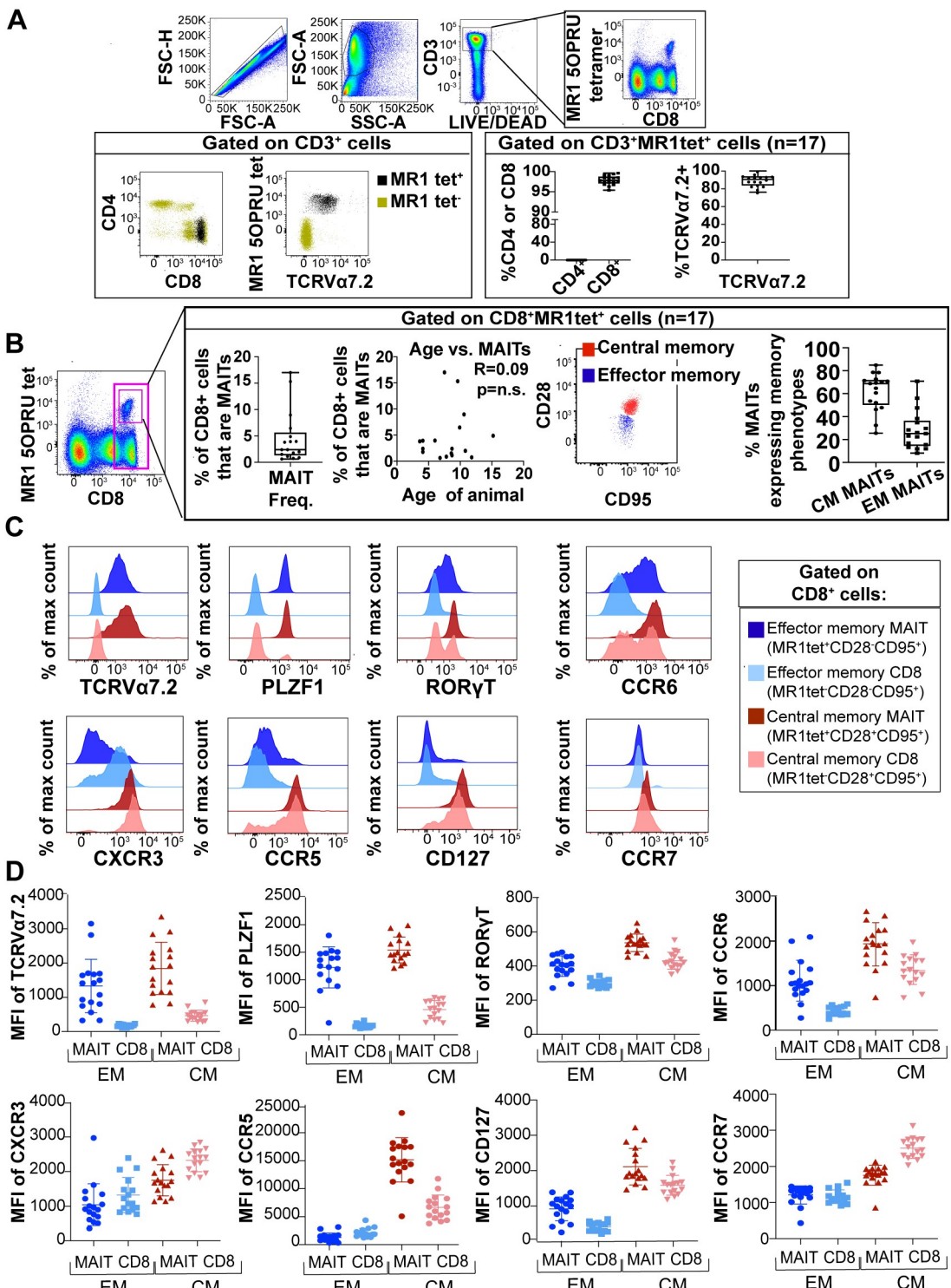

**Fig 1. Characterization of MAIT cells in SIV-naïve MCM.** A, PBMC from 17 SIV naïve MCM were stained with the MR1-5OP-RU tetramer along with antibodies to CD3, CD4, CD8, TCRVα7.2, CD28, CD95, PLZF1, RORγT, CCR6, CXCR3, CCR5, CD127, and CCR7. Flow cytometry was performed as indicated in Methods. Shown is a representative gating schematic for CD4, CD8, and TCRVα7.2 on CD3+MR1tet+ (black dots) and CD3+MR1tet- (gold dots) cells (left box). The right box shows the CD4, CD8, and TCRVα7.2 expression, as well as the frequency of CD3+MR1tet+ cells for all 17 MCM. B, MR1tet+ cells were characterized for their frequency as a percentage of the CD8+ cells, as well as their central memory (CD28+CD95+, red dots) or effector memory (CD28-CD95+, blue dots) phenotypes. CD8+MR1tet+ cells were also plotted against the age of each animal to

determine the relationship between MAIT frequencies and age. N.S. = not significant. C and D, The indicated EM and CM MAIT and conventional CD8 T cell populations were characterized for their expression of the indicated markers. Figure C shows a representative sample; Figure D shows the overall mean expression of the indicated markers across the population.

CD8+ T cell population, with an average of 4.6% of all CD8+ cells being MR1tet+ (Fig 1B). We found that of those cells, 61% ±16% of circulating MAIT cells expressed a central memory (CM) phenotype (CD28+CD95+) and that 28% ±17% expressed an effector memory (EM) phenotype (CD28-CD95+; Fig 1B), indicating there was extensive animal to animal variability. There were no correlations between the frequency of MAIT cells in the peripheral blood and the age of the animal (R = 0.09, Fig 1B).

We also examined the expression of cell surface markers and transcription factors that are associated with MAIT cells, such as CCR6 [37], PLZF1 [38], and RORγT [37, 39, 40]. A representative example of the expression of these markers by CM and EM MAIT cells (CD8+-MR1tet+ cells) compared to bulk CD8+ T cells (CD8+MR1tet- cells) is shown in Fig 1C. The mean fluorescence intensities (MFI) for each of these markers across all MCM that were studied are shown in Fig 1D. Similar to previous studies [13, 41, 42], MAIT cells, but not conventional CD8+ T cells, expressed TCRVα7.2 and PLZF1 (Fig 1C and 1D). Also consistent with previous studies (reviewed in [43]), CM MAIT cells expressed high levels of RORγT and CCR6, whereas the expression of these two markers was lower for EM MAIT cells, as well as both CM and EM bulk CD8+ T cells (Fig 1C and 1D).

We examined the expression of CXCR3 and CCR5, both of which are associated with the trafficking of T cells to the lungs [44]. CM MAIT cells expressed higher levels of CXCR3 and CCR5, when compared to EM MAIT cells (Fig 1C and 1D). We also measured expression of the LN homing marker CCR7 and the IL-7 receptor CD127 on MAIT and CD8+ T cells. Similar to other studies [45], we found that both CCR7 and CD127 were more highly expressed on CM MAIT and CD8+ T cells than on EM MAIT and CD8+ T cells (Fig 1C and 1D).

## MAIT cell numbers are not reduced during 6 months of SIV infection

Several cross-sectional studies reported fewer MAIT cells in HIV+ or SIV+ individuals, when compared to healthy controls [17, 18, 24, 25, 27]. In contrast, two recent longitudinal studies showed no decline in MAIT cell frequencies in SIV+ or SHIV+ pigtailed macaques and HIV + humans during the first year of infection [28, 29]. Here, we infected 4 MCM intrarectally with 3,000 TCID$_{50}$ of SIVmac239 and followed them for 6 months to determine if MAIT cells were depleted in the blood. (Fig 2A). Plasma SIV viremia is shown in Fig 2B.

The frequency and phenotype of peripheral MAIT cells were measured longitudinally in these 4 SIV+ animals (Fig 3A). Due to fluctuations in MAIT cell frequencies prior to SIV infection (Fig 3B) and the small number of animals, there were no statistically significant trends in total, CM, or EM MAIT cell frequencies in blood during the 6 months of SIV infection (Fig 3B), similar to what was observed in [29].

We then characterized the phenotypes of CM and EM MAIT cells during SIV infection using markers for T cell activation, proliferation, and exhaustion (CD69, CD25, HLADR, ki67, CD39, PD-1, TIGIT), trafficking (CCR6, CCR5, CCR7, CXCR3, CD127), and transcriptional markers of differentiation (T-bet, Eomes, RORγT, Fox3P) (Table 1, Fig 3A). For CM MAIT cells, we found transient increases in ki67, CD69, CD39, T-bet, and RORγT between days 14–21 post SIV (Fig 3C, top panels). In contrast, EM MAIT cells had a statistically significant increase in only ki67 (Fig 3D, bottom panels) between days 14–21. While the activation markers CD69 and CD39 also showed transient increases, these changes did not reach the level of significance (Fig 3D). Overall, these data suggest that MAIT cells are transiently activated

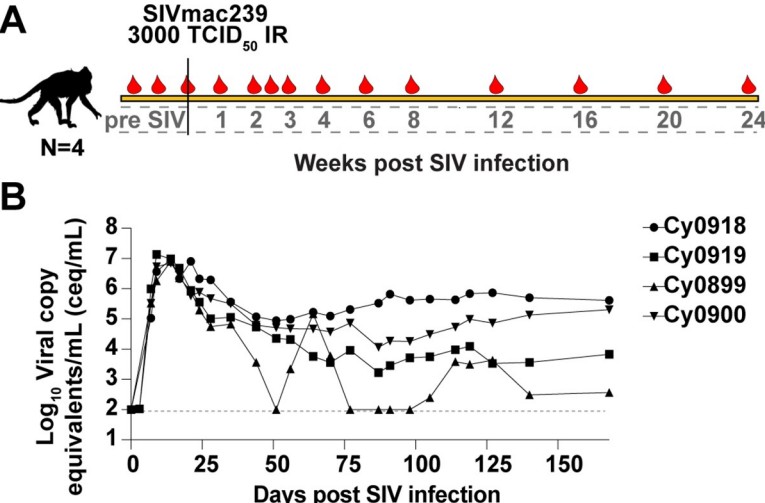

**Fig 2. SIVmac239 infection of MCM.** A, Four MCM were infected intrarectally with 3000 TCID50 of SIVmac239. Blood was collected at the times post-SIV infection depicted by the red droplets. B, Plasma viral loads were determined as indicated in the Methods. The Log$_{10}$ virus copy equivalents/mL (ceq/mL) were graphed for each time point. The limit of detection was $1 \times 10^2$ ceq/mL, as indicated by the dotted line.

during acute SIV infection (days 14–21 post infection), but then returned to basal activation levels. We did not assess if the activation of MAIT cells or their increased expression of ki67 was occurring in a TCRVα7.2-dependent or -independent manner, but it was assumed to be the latter based on previous studies of MAIT cell activation during viral infections (reviewed in [46]).

## SIV infection does not impair circulating MAIT cell frequency or phenotype during Mtb infection

The role of MAIT cells within granulomas of Mtb-infected individuals is not well established and whether SIV co-infection alters the frequency and function of MAIT cells within Mtb-affected tissues is unknown. Here, we used our SIV/Mtb co-infection model [33] to determine whether SIV disrupts the response of MAIT cells to infection with Mtb. We bronchoscopically infected 11 MCM with a low dose of Mtb Erdman strain. Eight other MCM were first infected intrarectally with 3,000 TCID$_{50}$ SIV for 6 months and then co-infected with the same dose and route of Mtb as the Mtb-only group (Fig 4A). Blood was collected at the timepoints indicated by the red droplets in the diagram (Fig 4A).

We examined the frequency of circulating MAIT cells within the two cohorts (Fig 4B, left panel). Similar to the animals infected with SIV alone (Fig 3), chronic SIV infection did not lead to differences in the absolute number of circulating MAIT cells by 6 months of SIV infection (Fig 4B, left panel; red dots). The absolute number of MAIT cells also remained unchanged for both the SIV-naïve and SIV+ groups following infection with Mtb (Fig 4B, left panel). Additionally, no changes were observed in the absolute numbers of CM or EM subpopulations of MAIT cells (Fig 4B, middle and right panels).

Phenotypic changes in the expression of surface markers and transcription factors associated with T cell activation, proliferation, differentiation, and exhaustion were determined as described in Fig 3A. While the expression of most phenotypic markers remained unchanged for both cohorts, there were statistically significant increases in the frequency of circulating MAIT cells expressing ki67 following Mtb infection in both SIV-naïve and SIV+ animals (Fig

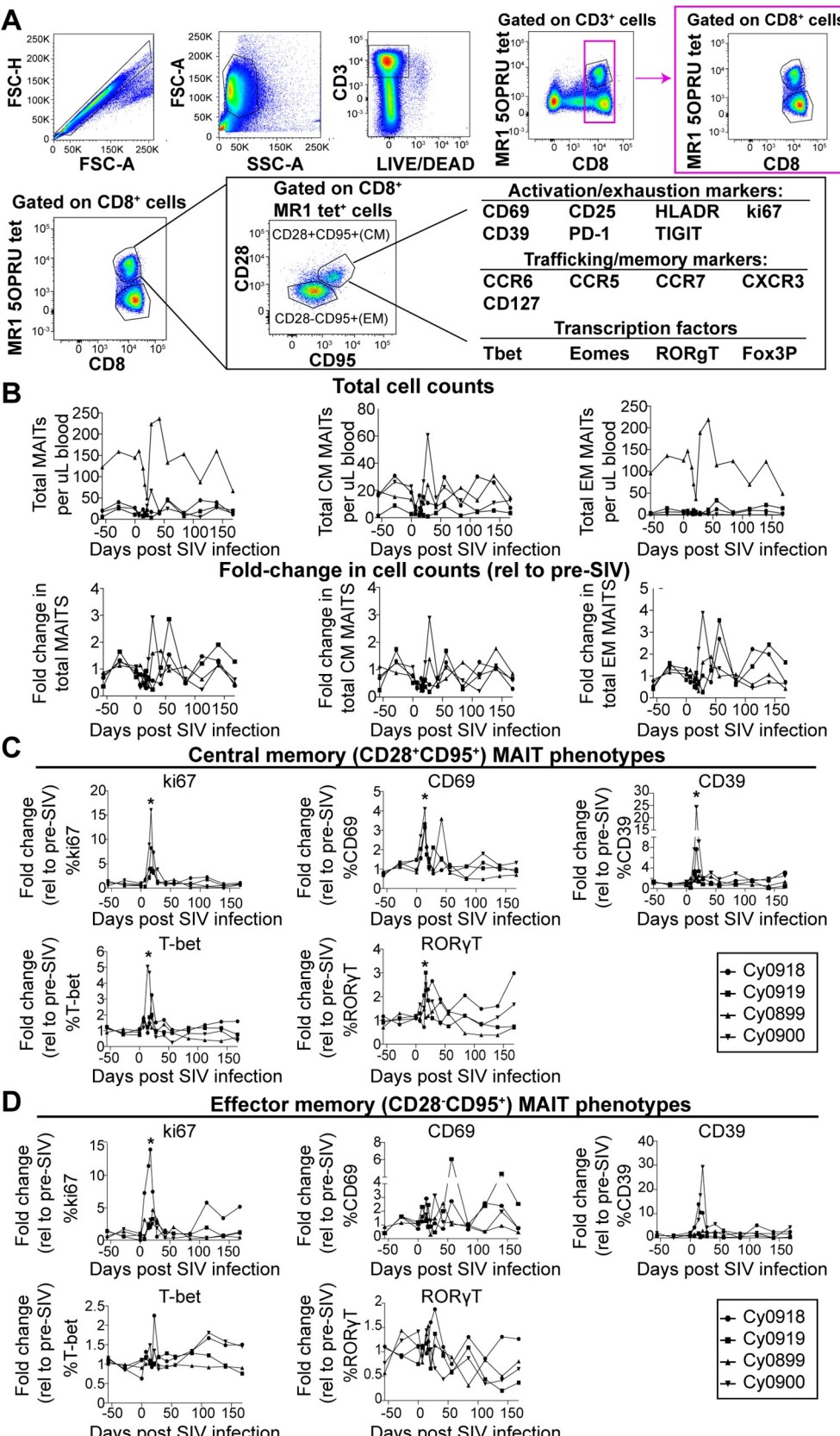

**Fig 3. MAIT cells in blood express an activated phenotype during acute SIV infection, but do not change in frequency longitudinally.** A, Flow cytometry was performed on cryopreserved PBMC from the timepoints indicated in Fig 2A. Shown is the gating schematic used to examine the expression of the indicated markers on the EM and CM subpopulations within the CD8+MR1tet+ parent gate. B, Total CM and EM MAIT cell counts (top panels) were determined for each animal for each timepoint post-SIV infection. The fold-change in cell counts (bottom panels) was determined by dividing the total cell counts for each timepoint by the average of the total cell counts pre-SIV infection. C and D. Flow cytometry was performed as described in (A) and the fold change in the percent of CM (C) or EM (D) MAIT cells expressing the indicated markers relative to their pre-SIV expression levels were determined. For statistical analysis, repeated measures non-parametric ANOVA tests with Tukey's multiple test correction were used to determine statistical significance. * p<0.05.

4C, left top panels). There were no marked differences in ki67 expression by MAIT cells between the two cohorts for any time point post-Mtb infection (Fig 4C, left bottom panel). Interestingly, the frequency of circulating MAIT cells expressing the early activation marker CD69 trended significantly higher following Mtb co-infection in SIV+ animals, but not in the SIV-naïve cohort (Fig 4C, right top panels). The frequency of CD69+ MAIT cells at 6 weeks post-Mtb infection was significantly higher in SIV+ MCM, compared to animals that were SIV-naïve (Fig 4C, right bottom panel).

**Table 1. Antibodies used in staining panels for flow cytometry.**

| Marker | Clone(s) | Fluorochrome(s) | Purpose of Use | Surface/Intracellular/intranuclear |
|---|---|---|---|---|
| CD45 | D058-1283 | BV786 | Lineage | Surface |
| CD3 | SP34-2 | AF700, BV650 | T cell marker | Surface |
| CD4 | OKT4, L200 | PE Cy7, BV510, BV711, BV786 | T cell marker | Surface |
| CD8 | RPAT8, SK1 | PacBlue, BV510, BV605, BV711, BV786 | T cell marker | Surface |
| CD206 | 19.2 | PE Cy5 | Macrophage exclusion | Surface |
| TCRVα7.2 | 3C10 | BV421, BV605 | MAIT cell marker | Surface |
| CCR5 | J418F1 | BV421 | Chemokine/MAIT lung homing | Surface |
| CCR6 | 11A9 | PE CF594, PE Cy7 | Chemokine/gut homing | Surface |
| CD127 | MB15-18C9 | PE | IL-7 receptor | Surface |
| CXCR3 | G025H7 | PE Dazzle 594 | Chemokine/MAIT lung homing | Surface |
| CCR7 | FAB197F | FITC | Chemokine/lymph homing | Surface |
| CD28 | CD28.2 | APC, BV510 | T cell memory | Surface |
| CD95 | DX2 | PE Cy5 | T cell memory | Surface |
| T-Bet | 4B10 | BV605 | Transcriptional marker | Intranuclear |
| Eomes | WD1928 | PE Cy7 | Transcriptional marker | Intranuclear |
| RORγT | AFKJS-9 | PE | Transcriptional marker | Intranuclear |
| PLZF1 | R17-809 | PE CF594 | Transcriptional marker | Intranuclear |
| Fox3P | 150D | AF488, AF647 | Transcriptional marker | Intranuclear |
| HLADR | G46-6 | BV650 | T cell activation | Surface |
| CD39 | eBioA1 (A1) | FITC, PE Cy7 | T cell activation | Surface |
| CD25 | BC96, M-A251 | PE, BV605 | T cell activation | Surface |
| CD69 | TP1.55.3 | ECD | T cell activation | Surface |
| ki67 | B56 | AF647 | T cell proliferation | Intracellular |
| TIGIT | MBSA43 | FITC, PerCP EFluor710 | T cell activation/exhaustion | Surface |
| PD1 | EH12.2H7 | BV605 | T cell activation/exhaustion | Surface |
| IFNγ | 4S.B3 | FITC, BV510 | Cytokine | Intracellular |
| TNFα | Mab11 | AF700, PerCP Cy5.5 | Cytokine | Intracellular |
| CD107a | H4A3 | APC, BV605 | Degranulation | Surface |
| LIVE/DEAD | ---- | Near IR | Dead cell stain | ---- |

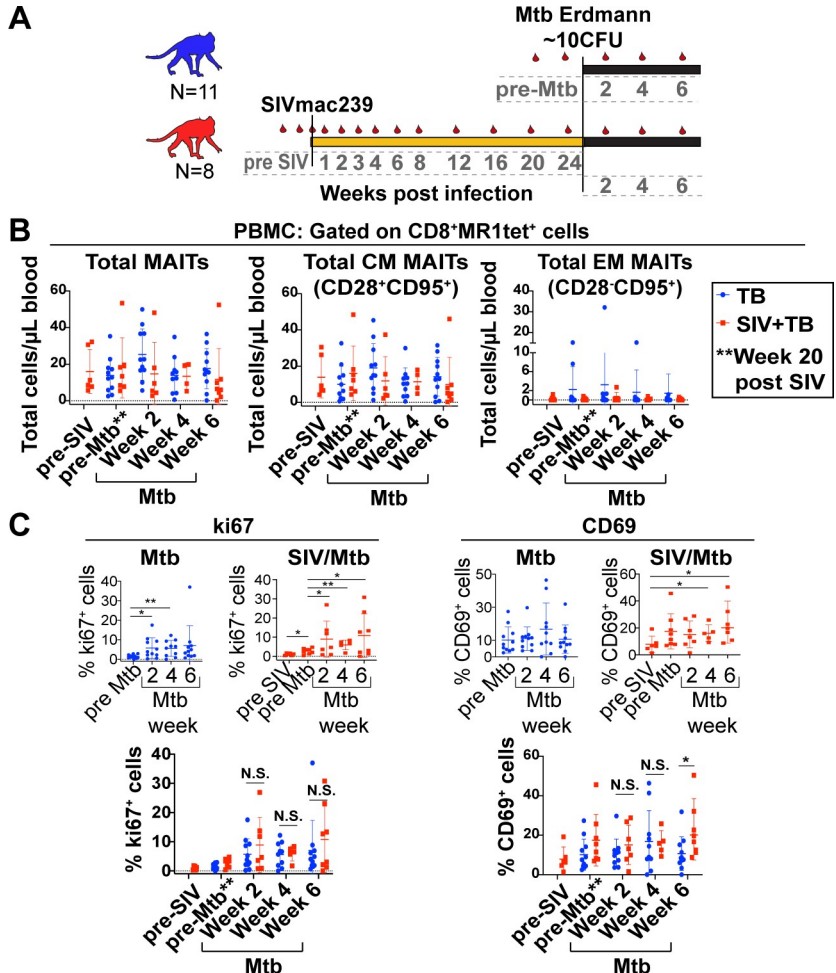

**Fig 4. MAIT cells in blood upregulate ki67 during Mtb infection, but do not increase in frequency during Mtb infection of SIV-naïve or SIV+ MCM.** A, Schematic of the study design to investigate the role of MAIT cells during Mtb infection in SIV-naïve (blue, n = 11) or SIV+ (red, n = 8) MCM. Red droplets indicate blood collection timepoints. B, PBMC from the indicated time points post-SIV and/or post-Mtb infection were stained for MAIT cells, and flow cytometry was performed as described in Fig 3A. Total, CM, and EM MAITs for SIV-naïve MCM (blue dots) or SIV+ MCM (red dots) were calculated. C, Flow cytometry was performed as described in Fig 3A to measure the expression of the indicated phenotypic markers on MAIT cells. The expression of ki67 and CD69 on MAIT cells was measured longitudinally for each cohort (top panels). The mean +/-SEM are shown for each timepoint, and Wilcoxon Rank signed tests for matched pairs were performed to determine statistical significance comparing pre-infection timepoints to post-infection timepoints; * p<0.05, ** p<0.005. Mann-Whitney tests were used to compare differences between SIV-naïve (blue dots) and SIV+ (red dots) MCM for the same timepoints post-Mtb infection; * p<0.05. N.S. = not significant.

## Characterization of MAIT cells in BAL during SIV and/or Mtb infection

BAL fluid can be collected longitudinally following Mtb infection in macaques to assess the early immune response within the airways during SIV/SHIV and Mtb infection [22, 29, 47]. We collected longitudinal BAL samples during SIV infection and characterized the frequency and phenotype of MAIT cells as indicated in Fig 5A. At 4 weeks after SIV infection, there was an increase in the frequency of CD45+ lymphocytes within the airways, which remained elevated for the rest of the study (Fig 5B, left panel). Concurrently, the percent of total airway CD8+ cells that were MR1 tet+ decreased (Fig 5B, middle panel). This was likely related to an influx of CD3+CD8+MR1 tet-negative cells (Fig 5B right panel), as previously reported for

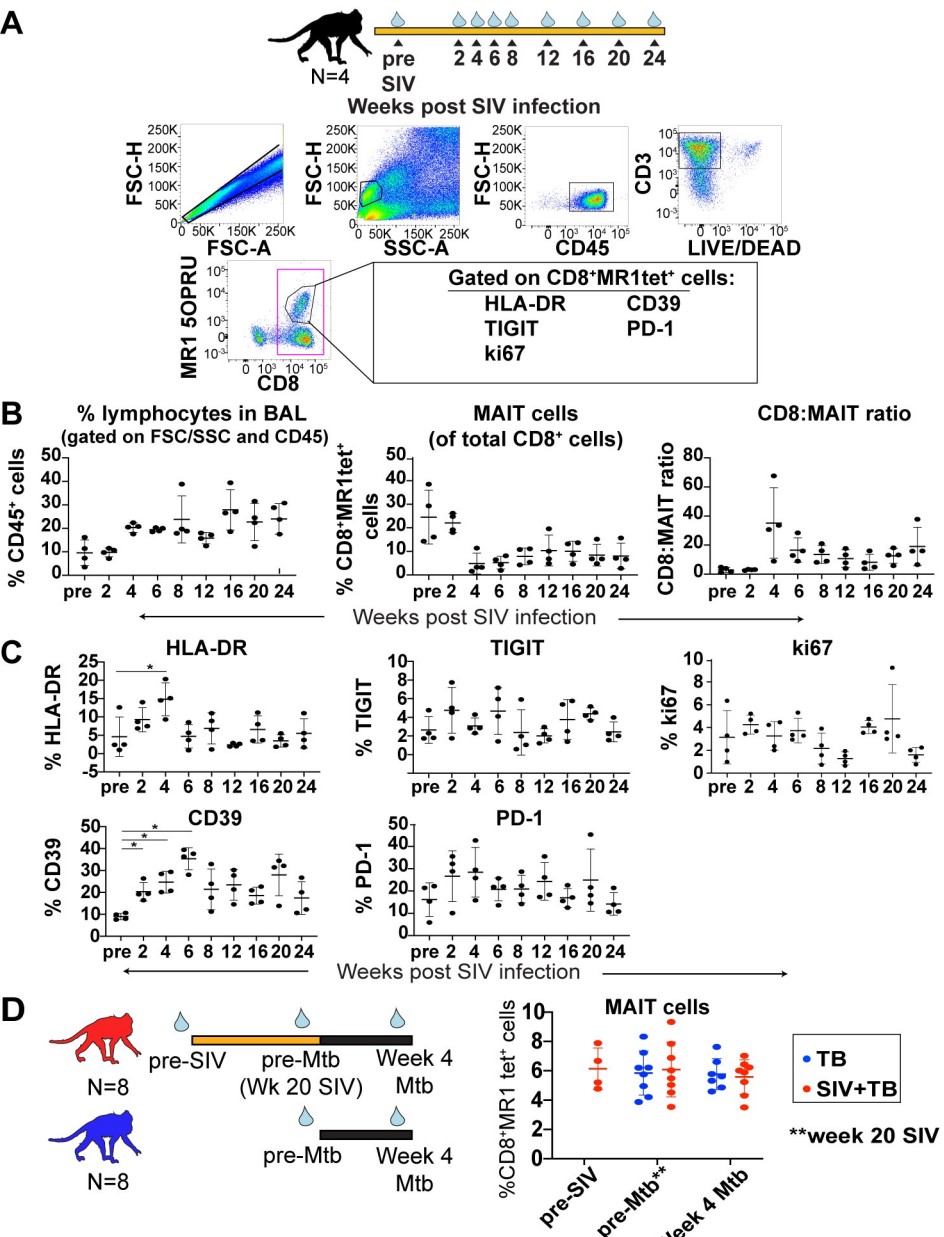

**Fig 5. Longitudinal analysis of MAIT cells in BAL.** A, BAL fluid from SIVmac239-infected MCM was collected at the indicated time post-SIV infection. Flow cytometry was performed, and a representative image of the gating schematic used to determine MAIT cell frequencies within BAL is shown. B, The percent of lymphocytes present within each BAL sample for each indicated timepoint was determined (left panel). Within the CD3+ parent gate, the frequency of MR1tet+ cells (middle panel) and the CD8:MAIT cell ratio (right panel) were determined for each timepoint. C, The frequency of CD8+MR1tet+ cells expressing HLA-DR, TIGIT, ki67, CD39, and PD-1 was determined. The mean +/-SEM are shown for each timepoint, and repeated measures non-parametric ANOVA tests with Tukey's multiple test correction were performed to determine statistical significance between timepoints; * $p \leq 0.05$. D, BAL was collected at the indicated timepoints (blue droplets) pre- and post-SIV and/or Mtb infection for 8 SIV/Mtb coinfected MCM (red) and 8 Mtb-infected MCM (blue). Flow cytometry was performed as described in (A) and the frequency of CD3+MR1tet+ cells was determined for the indicated timepoints.

SIV infection [47–49], thus increasing the ratio of conventional CD8+ T cells to MAIT cells (Fig 5B).

Similar to the activation of circulating MAIT cells during SIV infection (Fig 3), we observed significant increases in MAIT cells expressing HLA-DR and CD39 in the BAL, peaking between 2–6 weeks after SIV infection (Fig 5C, left-most panels). There were no changes during SIV infection in the frequency of MAIT cells expressing activation/exhaustion markers such as TIGIT and PD-1, or the proliferation marker, ki67 (Fig 5C, middle and right panels). We also compared the expression of HLA-DR, CD39, TIGIT, PD-1, and ki67 on MAIT cells within circulating PBMC collected at the same timepoints as the BAL (S1 Fig). Similar studies have been performed in HIV+ individuals to compare the phenotypes of conventional CD4 + and CD8+ T cells in the BAL vs PBMC [50]. Regardless of the timepoint or infection status, HLA-DR, CD39, TIGIT, and PD-1 were more highly expressed on MAIT cells in the BAL compared to the PBMC (S1 Fig), consistent with some of the findings with conventional T cells in HIV+ individuals [51]. The MAIT cells in the BAL expressed lower ki67 levels than those in the PBMC, again consistent with the findings in HIV+ individuals [51].

We also collected BAL fluid from SIV+ and SIV-naïve animals after Mtb infection and performed flow cytometry (Fig 5D). We found that Mtb infection did not affect the frequency of MAIT cells in either the SIV+ or SIV-naïve cohorts (Fig 5D). Due to small numbers of MAIT cells within the BAL of the animals within these cohorts, we were unable to confidently characterize the expression of activation markers.

## Characterization of MAIT cells in tissues

To assess whether a pre-existing SIV infection impairs MAIT cell frequency or function within Mtb-affected tissues, LNs and granulomas were collected from SIV-naïve or SIV+ MCM 6 weeks after Mtb infection and flow cytometry was performed (Fig 6A). At this relatively early time point post Mtb infection (or co-infection), the TB disease (e.g. pathology, bacterial load) did not differ significantly between the two cohorts.

The frequency of MAIT cells within Mtb-affected tissues was quantified for each cohort (Fig 6B and 6C). Phenotypic differences in the expression of PD-1, TIGIT, ki67, IFNγ, and TNFα were measured on MAIT cells residing within LNs and Mtb-affected tissues and compared across the relevant cohorts.

We found that within Mtb-affected LNs, there was an increase in both the frequency and absolute numbers of MAIT cells in SIV/Mtb co-infected MCM compared to MCM infected with Mtb alone (Fig 6B). This increase in MAIT cell frequency appeared to be related to SIV infection, as MAIT cells were also at a higher frequency in the peripheral LN of animals infected with SIV alone when compared to the SIV-naïve, Mtb infected group (Fig 6B). We found a weak negative correlation between MAIT cell frequency and bacterial burden in the SIV/Mtb co-infected group (R = -0.46, p = 0.002; Fig 6B, right panels), but this correlation was not significant for the SIV-naïve, Mtb infected group. MAIT cells in the LN of SIV+ animals, regardless of whether the animals were co-infected with Mtb, expressed higher levels of PD-1 and TIGIT compared to SIV-naïve animals (Fig 6B, bottom left panels). While the frequency of MAIT cells producing IFNγ was not different between cohorts, we did observe a reduction in the frequency of MAIT cells producing TNFα in SIV/Mtb co-infected MCM compared to MCM infected with Mtb alone (Fig 6B, bottom right panel). Overall, we conclude that SIV infection leads to an increase in the frequency of activated MAIT cells in LNs.

Similar comparisons were made for the cells isolated from granulomas excised from SIV + and SIV-naïve animals following Mtb infection. The percentage of MAIT cells within the granulomas was lower in the SIV+ cohort, compared to the SIV-naïve cohort (Fig 6C, top left panel). However, absolute MAIT cell counts remained similar between both cohorts (Fig 6C, top middle panel). This suggests that an influx of non-MAIT CD8+ T cells into granulomas

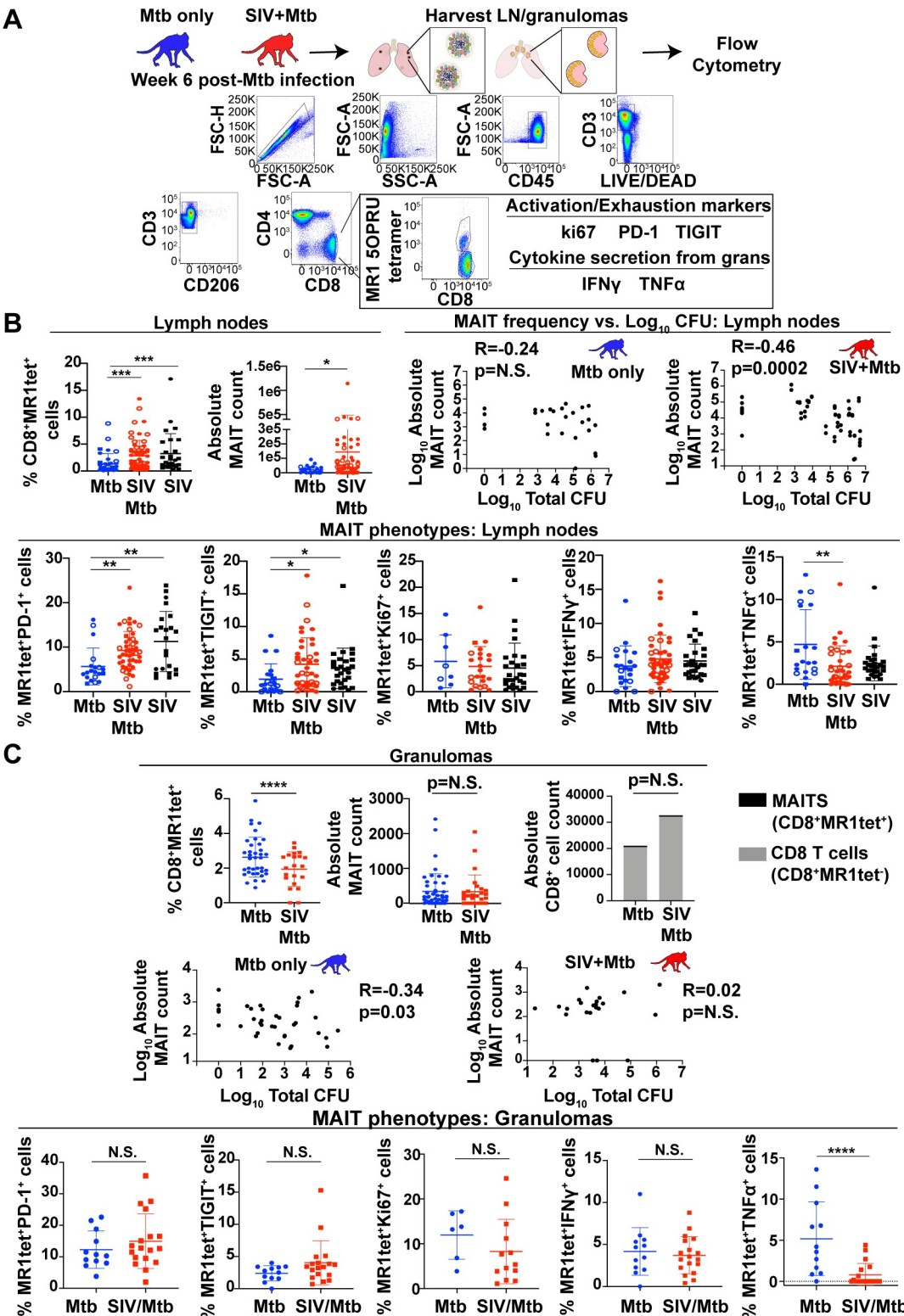

**Fig 6. Analysis of MAIT cell frequency in Mtb-infected tissues.** A, Mtb-infected tissues were collected from SIV-naïve and SIV+ MCM 6 weeks after Mtb infection. Tissues were processed according to Methods and cell homogenates were used for flow cytometric analysis. Shown is the gating schematic used to examine the expression of the indicated markers on the CD8 +MR1tet+ cells. B, Homogenates collected from thoracic (circles) and extrathoracic (squares) LNs present in MCM infected with Mtb (blue), SIV/Mtb (red), or SIV alone (black) were analyzed as indicated in (A) for the frequency of CD8+MR1tet

+ cells. Open circles/squares indicate samples from animals infected with Mtb that were sterile (Mtb culture negative). The mean +/-SEM are shown for each group, and Kruskall-Wallis tests with Dunn's adjusted p values were performed to calculate statistical significance; *** p<0.0001; * p<0.05. Top right two panels: Pearson correlation coefficients were calculated to determine the relationship between the log-transformed values of the absolute MAIT cell counts and the log-transformed total CFU for each sample. Shown are the correlations for samples collected from SIV-naïve (blue) and SIV+ (red). Bottom panels: the expression of the indicated markers was determined for CD8+MR1tet+ cells. Prior to statistical analysis, the mean expression of the indicated markers per animal were graphed and determined to eliminate animal bias. The mean +/-SEM are shown for each group, with Kruskall-Wallis tests with Dunn's adjusted p values were performed as indicated above for statistical significance; ** p<0.001, * p<0.01. C, Top panels: CD8+MR1tet+ frequencies (left), and absolute cell counts (middle) for granulomas of Mtb-infected MCM (blue) and SIV/Mtb-infected MCM (red) were determined as in B. Top right; absolute CD8+ T cell (CD8+MR1tet-; grey bars) and MAIT cell (CD8+MR1tet+, black bars) counts within granulomas were determined as described in (B). Middle panels: Pearson correlation coefficients were determined for the relationship between total MAIT cell counts and total CFU as indicated in (C). Bottom panels: MAIT cells (CD8+tet+ cells) within granulomas were characterized for their expression of the indicated markers as described in (B). Prior to statistical analysis, the mean expression of the indicated markers per animal were graphed and determined to eliminate animal bias. The mean +/-SEM are shown for each group, and Mann-Whitney statistical tests were performed; **** p<0.00005.

(Fig 6C, top right panel) decreased the frequency of MAIT cells in those lesions. While there was a weak negative correlation between absolute MAIT cell counts and bacterial burden in granulomas from the SIV-naïve cohort (Fig 6C, middle left graph), there was no such correlation in the SIV+ cohort (Fig 6C, middle right graph). In contrast to the LN, there were no differences in the frequencies of MAIT cells expressing PD-1, TIGIT, ki67, or IFNγ (Fig 6C) in the granulomas of SIV+ animals compared to those that were SIV-naïve. Notably, there was a significant decrease in the frequency of MAIT cells producing TNFα in the SIV+ cohort compared to the SIV-naïve cohort (Fig 6C, bottom right panel). Overall, the frequency of MAIT cells did not appear to correlate with bacterial burden from both SIV-naïve and SIV+ animals. However, at the sites of Mtb infection, there was a decrease in TNFα production by MAIT cells in SIV+ MCM compared to those that were SIV-naive.

### *Ex vivo* assessment of MAIT cell function throughout the course of SIV infection

MAIT cell function can be assessed *ex vivo* by using paraformaldehyde-fixed bacteria as a stimulus [52]. We stimulated MAIT cells with both paraformaldehyde fixed *E. coli* or *M. smegmatis* as the MR1 molecule presents a different repertoire of ligands derived from each bacterial species [53]. PBMC from SIV-naïve MCM were incubated with 10 CFU/cell of fixed *E. coli* or *M. smegmatis* and flow cytometry was used to measure cytokine production and degranulation of MAIT cells (Fig 7A). Examples of the gating schematic used for flow cytometry and production of TNFα, IFNγ, and CD107a by MAIT cells can be found in S2 Fig. Among these samples, incubation of PBMC with *E. coli* induced a greater production of TNFα and IFNγ, as well as increased surface expression of the degranulation marker CD107a on MAIT cells (Fig 7A) when compared to stimulation with *M. smegmatis*.

We wanted to determine whether the weaker stimulation of MAIT cells by ligands from *M. smegmatis*, compared to *E. coli*, ligands was related to differential uptake of the bacteria by antigen presenting cells. We labeled equal amounts of *M. smegmatis* and *E. coli* with a pHRODO green dye which increases in fluorescence intensity when labeled bacteria are taken up by antigen presenting cells, and has been used previously to measure the uptake of fixed *E. coli* by antigen presenting cells for MAIT cell functional assays [52]. Increasing numbers of labeled bacteria were delivered to SIV-naïve PBMC and the fluorescence intensity was measured after 3 hours. For each dose of labeled bacteria tested, there were no significant differences in the ability of PBMC to uptake *M. smegmatis* compared to *E. coli* (Fig 7A, right panel).

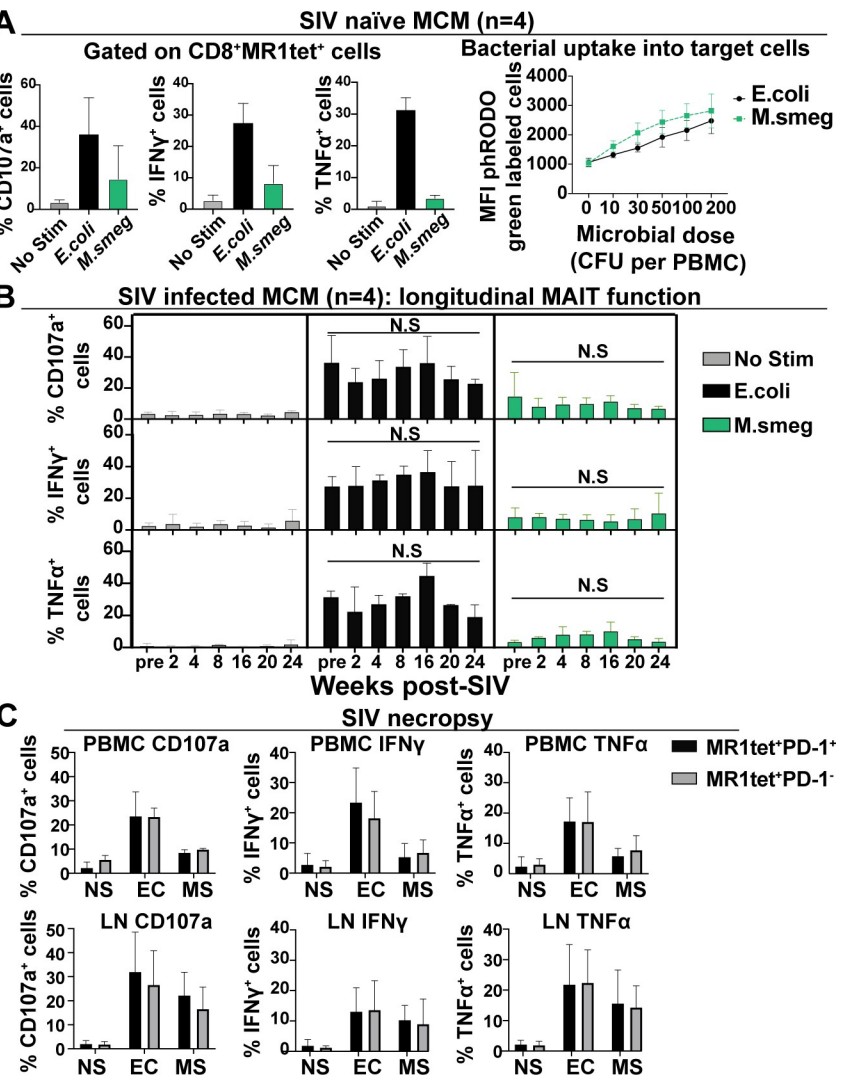

**Fig 7. Longitudinal SIV infection does not impair *ex vivo* MAIT cell function.** A, Functional assays were performed as described in Methods using PBMC from four SIV-naïve MCM. Cells were stimulated for 16 hours with either 10 CFU per cell of fixed *E. coli* (black bars) or *M. smegmatis* (green bars). The samples were stained as indicated in Methods, and the expression of CD107a, IFNγ, and TNFα were measured on CD8+MR1tet+ cells from stimulated samples compared to an unstimulated sample control. Right panel: Bacterial uptake into PBMC was measured by labeling equal amounts of bacteria with phRODO green dye as indicated in Methods, then incubating increasing microbial doses of the labeled bacteria with PBMC for 5 hours. B, Functional assays and flow cytometry were performed as described in (A) using PBMC collected at the indicated timepoints post-SIV infection for four SIV-infected MCM. To compare expression of the indicated cytokines longitudinally for each individual animal for a single stimulus, repeated measures ANOVA tests with Tukey's multiple test correction was performed. C, LNs and PBMC samples collected at the time of necropsy for the four SIVmac239 infected MCM described in (B) were used, and functional assays using fixed *E. coli* (EC) and *M. smegmatis* (MS), or no stimulation (NS) were performed as described in (A) and (B). MAIT cells were gated during flow cytometric analysis into two separate populations: CD8+MR1tet+PD-1+ (black bars) and CD8+MR1tet+PD-1- (gray bars). CD107a, IFNγ, or TNFα were measured as shown. Statistical analysis was performed as described in (A).

To determine if SIV infection alters MAIT cell function over time, PBMC were incubated with 10 CFU/cell of fixed *E. coli* or *M. smegmatis* (Fig 7B). We observed robust cytokine and CD107a production when cells were stimulated with *E. coli*, but weaker responses when *M. smegmatis* was used (Fig 7B). No significant differences were observed in the production of

TNFα, IFNγ, or CD107a at any time point measured pre- or post-SIV infection for either *E. coli* or *M. smegmatis* stimuli. Overall, these data suggest that SIV infection does not affect the function of circulating MAIT cells over time.

Functional assays were performed using PBMC and LN samples collected at the time of necropsy to determine if MAIT cell function was affected by the increased expression of PD-1 we observed during SIV infection. We found that in both PBMC and LN, cytokine production was similar for PD-1+ and PD-1-negative MAIT cells after stimulation with either *E. coli* or *M. smegmatis* (Fig 7C). Taken together, no defects in *ex vivo* MAIT cell activity were apparent during SIV infection in response to either bacterial stimulus.

### Reduced production of TNFα by MAIT cells after Mtb infection in SIV + MCM, compared to SIV-naïve MCM

To determine whether Mtb infection could enhance or inhibit the *in vitro* MAIT cell response to bacterial stimuli, when compared to naïve animals, we performed *in vitro* functional assays using PBMC from the SIV-naïve and SIV+ animals following Mtb infection (Fig 8). Although stimulation with *E. coli* elicited more robust cytokine production when compared to stimulation with *M. smegmatis* (Fig 8A), there were no differences in MAIT cell cytokine production before and after Mtb infection. Likewise, for the SIV+ MCM, MAIT cell function did not differ before and after SIV infection (Fig 8B). However, we found a lower frequency of MAIT cells producing TNFα after Mtb co-infection in response to both the *E. coli* and *M. smegmatis* stimuli (Fig 8B, right panel). This is consistent with the reduced production of TNFα by MAIT cells we observed directly *ex vivo* from LNs and granulomas from SIV/Mtb co-infected MCM when compared to animals infected with Mtb alone (Fig 6). We analyzed cytokine production within the same *in vitro* functional assays for the CD8+MR1tet- cells (conventional CD8 T cells, S3A–S3C Fig). Fixed bacteria were able to stimulate the production of TNFα and IFNγ, but not CD107a, in conventional CD8+ T cells (S3A and S3B Fig). There were no differences longitudinally in cytokine production by conventional CD8+ T cells for the MCM infected with Mtb alone (S3A Fig). However, there were impairments in both TNFα and IFNγ production longitudinally in the SIV+ MCM, particularly by 6 weeks post Mtb infection (S3B Fig). Further analysis of conventional CD4 and CD8 T cells is currently underway. While fixed bacteria are not the ideal stimulus for conventional CD8+ T cells, these observations further strengthen the idea that SIV infection may indirectly impair the antibacterial immune response by inhibiting cytokine production of other immune cell subsets.

### Discussion

We previously observed that rapid early dissemination of granulomas was common among SIV+ macaques between 4 and 8 weeks after Mtb infection, leading to their rapid TB progression and mortality [33]. Here, we wanted to test the hypothesis that SIV-dependent dysregulation of MAIT cells could impair their frequency and function within Mtb-affected tissues. A key aspect of our study was that we performed necropsies at 6 weeks after Mtb challenge. Thus, we captured the phenotypes of the host immune response, right when Mtb control in the SIV+ and SIV-naïve animals was beginning to diverge, and before they were influenced by advanced TB disease. While there were no significant differences in measures of TB disease (e.g. pathology, PET/CT imaging, bacterial load) between the SIV+ or SIV-naïve MCM at the time the animals were necropsied, we would expect that the SIV+ animals would have rapidly progressed to advanced TB disease [33]. Thus, defects observed in the immune responses in the SIV+ animals, when compared to those that were SIV-naïve, should be considered as potential mechanisms for increased TB progression in the SIV/HIV population.

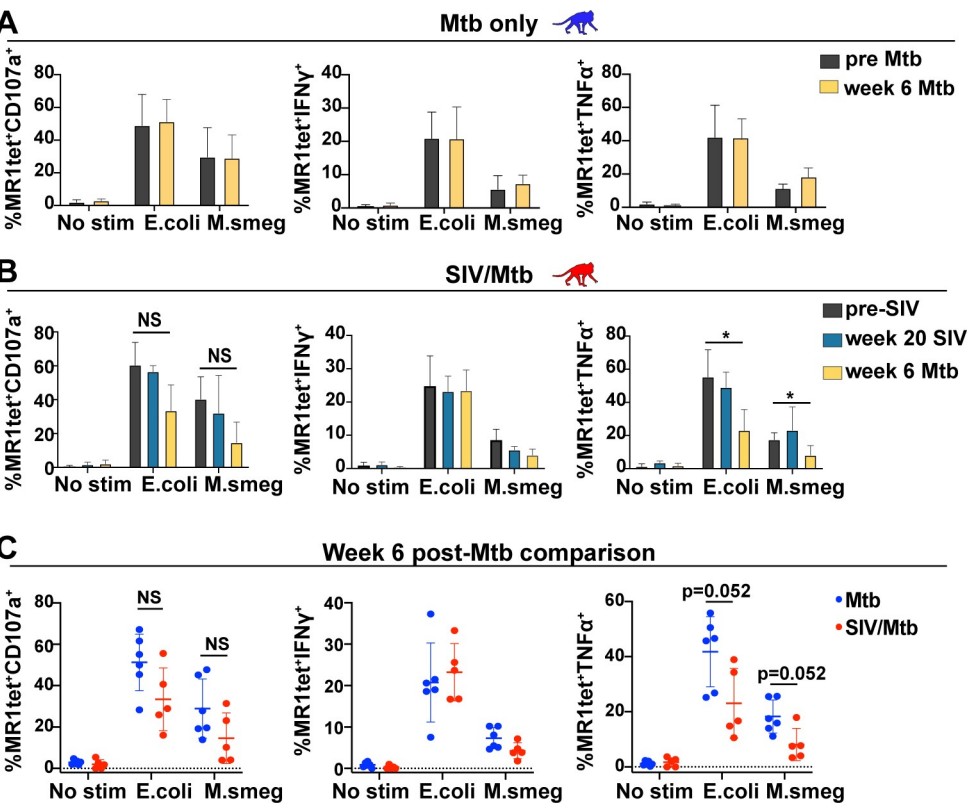

**Fig 8. SIV and Mtb co-infection results in functional impairment of TNFα production in MAIT cells in the blood.**
A, Functional assays were performed as described in Fig 7A using PBMC from pre-Mtb (dark grey) and 6 weeks post-Mtb (yellow) infection from seven Mtb-infected MCM (blue monkeys). Wilcoxon Rank sum statistical tests for matched pairs were performed to compare the expression of the indicated cytokines or CD107a longitudinally for a single stimulus. B, Functional assays were performed as decribed in (A) using PBMC from pre-SIV (dark grey), 20 weeks post-SIV (teal), and 6 weeks post-Mtb (yellow) infection from 6 SIV/Mtb coinfected MCM. Wilcoxon Rank sum statistical tests for matched pairs were performed to compare the expression of the indicated cytokines and CD107a longitudinally for a single stimulus. C, The functional assays performed in (A) and (B) using PBMC from six weeks post-Mtb infection for SIV-naïve (blue dots) and SIV+ (red dots) MCM were plotted on the same graphs to compare the expression of CD107a (first panel), IFNγ (second panel), or TNFα (third panel) amongst the two cohorts. Mann-Whitney tests were performed to determine statistical significance.

In this manuscript, we specifically characterized MAIT cells in SIV, Mtb, and SIV/Mtb co-infected MCM to determine whether SIV infection dysregulated MAIT cell function. We phenotyped MAIT cells longitudinally during SIV infection, and also described the early MAIT cell response to Mtb infection within Mtb-affected tissues (e.g. granulomas and LNs). Finally, we show for the first time that SIV co-infection affects MAIT cell frequency and function within granulomas. Separate studies evaluated the impact of SIV infection on the conventional T cell response.

One key finding was that MAIT cells present in the granulomas and LNs of SIV/Mtb co-infected animals produced less TNFα when measured directly *ex vivo*, compared to those present in the granulomas of MCM infected with Mtb alone (Fig 6). The decreased TNFα production by MAIT cells found in granulomas was supported by *in vitro* assays showing a trend towards decreased TNFα production in MAIT cells isolated from SIV/Mtb co-infected animals when stimulated with either *E. coli* or *M. smegmatis* (Fig 8). Similar defects were observed in TNFα production from conventional CD8 T cells in the same in vitro assays (S3 Fig). Despite this, the decreased production of TNFα by MAIT cells alone is very unlikely to be the only

determinant for containment of Mtb infection. However, others have shown that anti-TNF therapy or blocking TNFα production can lead to exacerbation of primary TB or reactivation of latent Mtb infection in several animal models as well as Mtb-infected individuals [54–58]. Defective TNFα production has also been observed when alveolar macrophages were infected with HIV *in vitro* [59]. Furthermore, a recent study has shown that MAIT cells were defective in the production of TNFα during SHIV infection [60]. Overall, the mechanism by which SIV/HIV impairs TNFα production has not yet been fully determined, but our data suggests that SIV/HIV infection may dysregulate the ability of MAIT cells (Fig 8) as well as conventional CD8 T cells (S3 Fig) to produce TNFα in response to mycobacteria. What role this may play in Mtb containment has yet to be tested.

We were surprised to find that SIV infection did not dysregulate the activation and proliferative capacity of MAIT cells in the granulomas (Fig 6C). Overall, MAIT cells failed to robustly respond to Mtb infection within the tissues (Fig 6), similar to a study performed in SIV-naïve rhesus macaques that also showed that MAIT cells present in granulomas failed to produce markers associated with proliferation [22]. One possible explanation for the *in vivo* (Fig 6) findings reported here and by others [22] is that Mtb produces metabolites distinct from those produced by *E. coli* or *M. smegmatis*, and thus our *in vitro* assays missed the MAIT cells that specifically respond to Mtb. This is plausible, as different bacteria can produce distinct ligands for presentation by MR1 [53]. Furthermore, there are MAIT cells with distinct TCR sequences that can distinguish between ligands produced by *M. smegmatis* and Mtb [61]. Therefore, future studies using *in vitro* assays to characterize MAIT cells in Mtb-affected tissue may need to include Mtb as stimulating agent.

While we did find that *E.coli* and *M.smegmatis* differentially stimulated MAIT cells *in vitro*, there are caveats to these observations. It has been established that MAIT cells do not always respond to mycobacterial antigens by MR1/TCRVα7.2-dependent mechanisms [62]. For example, *Mycobacterium bovis* (BCG) can stimulate MAIT cells *in vitro* through TCRVα7.2-independent mechanisms [63]. On the other hand, MAIT cell clones derived from Mtb-infected patients have been observed to respond to Mtb in a manner dependent upon TCRVα7.2 and MR1 interaction [11]. There is some evidence that MAIT cell lines derived from human patients produce cytokines in response to *M.smegmatis* infected cells in an MR1-dependent manner [64]. It has yet to be tested whether the stimulation of MAIT cells by *M.smegmatis* is occurring in an MR1-dependent or -independent manner in our animal model. Furthermore, the fact that frozen PBMC were used for our functional assays should be noted, as it may affect the ability of antigen-presenting cells to effectively present metabolites or secrete cytokines. We were unable to obtain sufficient PBMC for the animals within this study to grow autologous monocytes as antigen presenting cells, but future studies could include them where possible.

While it is unclear whether MAIT cells have a direct role in antimycobacterial immunity, SIV co-infection clearly affects their ability to secrete certain cytokines, such as TNFα (Fig 8). MAIT cells secrete several proinflammatory cytokines in response to bacterial infection [43]. Cytokine secretion by MAIT cells responding to an Mtb infection could then provide signals to conventional T cells that either lead to their activation or trafficking to the lungs or other sites of infection [65, 66]. Furthermore, MAIT cells have recently been shown to provide signals to B cells to help develop antibody responses to pathogens [67]. Whether the SIV dependent dysregulation of MAIT cell cytokine production has an indirect effect on recruitment or activation of other immune cells during Mtb infection has not been assessed and may be interesting to pursue in future studies.

Another interesting observation was the phenotypic differences between MAIT cells within the LN versus those within granulomas. Granuloma-resident MAIT cells were similar between

the SIV-naïve and SIV+ cohorts, except for lower TNFα production in the latter group. In contrast, the MAIT cells within Mtb-affected LNs from SIV+ MCM displayed increased expression of PD-1 and TIGIT compared to SIV-naïve MCM (Fig 6). We hypothesize that the activation of MAIT cells in LNs was driven by SIV infection because both the SIV-only and SIV/Mtb co-infected MCM expressed higher levels of these markers (Fig 6B). While we did not find evidence that PD-1 expression was associated with MAIT cell exhaustion (Fig 7C), it is likely that ongoing SIV replication can lead to the activation of MAIT cells. This was also observed in the blood, where we detected transient increases in the circulating MAIT cells with a higher expression of ki67, CD69, CD39, RORγT, and T-bet, coincident with peak viremia (Fig 2, Fig 3B and 3C). The increase in ki67 is consistent with increased MAIT cell proliferation observed by others during acute SIV/HIV infection [28, 29]. This type of TCR-independent activation of MAIT cells has been observed for other viral infections as well [23]. We hypothesize that during active SIV replication, other immune cells are producing cytokines such as IL-12 and IL-18 that can activate MAIT cells [68–70]. Therefore, MAIT cell activation may be occurring indirectly though TCR-independent mechanisms, rather than directly through bacterial infection, but this hypothesis remains to be tested.

We did not explore the possibility that SIV infection might affect the trafficking of MAIT cells to the sites of Mtb infection via modulation of chemokine receptors. There was an increase in the number of MAIT cells present in the lymph nodes of SIV/Mtb co-infected MCM compared to the SIV-naïve cohort (Fig 6B, left panels). However, within the SIV/Mtb co-infected cohort we did observe an inverse relationship between the numbers of MAIT cells within Mtb-affected lymph nodes and the total CFU present in those lymph nodes (Fig 6B, right panels). We do not currently have evidence to support the hypothesis that MAIT cells had an enhanced mycobacterial killing capacity in the samples with low total CFU. However, it is possible that Mtb and/or SIV infection may affect the trafficking of MAIT cells to the lymph nodes. For example, there may be modulation of lymph node homing markers such as CCR7. Dendritic cells have been shown to upregulate CCR7 during Mtb infection [71]. Alternatively, increased MAIT cell activation during SIV and/or Mtb infection might increase the surface expression of gut homing markers, such as α4β7, as was shown during SIV infection [29]. Neither the modulation of CCR7 nor α4β7 on MAIT cells were evaluated in this study, but should be explored in future experiments in the SIV/Mtb co-infection model.

Overall, our study shows that SIV differentially dysregulates the phenotypes of MAIT cells, albeit depending on their tissue location. SIV infection activates MAIT cells and can impair their ability to produce TNFα in Mtb-affected tissues. This virus-dependent activation of MAIT cells has also been observed in HIV infection [28], and the failure of MAIT cells to respond in a robust way to Mtb has also been observed in the tissues of rhesus macaques [22]. In our study, we combine SIV with Mtb co-infection to show that that SIV can induce activation of MAIT cells *in vivo*, and can also impair their ability to produce TNFα upon co-infection with Mtb. This unique SIV/Mtb co-infection model provides the opportunity to dissect out how co-infection with two pathogens elicits distinct immune responses that may not be apparent when individuals are infected with only SIV or Mtb, and thus provide insights into the immunological defects that render HIV+ individuals so susceptible to TB.

## Materials and methods

### Animal care/Ethics statement

Mauritian cynomolgus macaques (*macaca fascicularis*; MCM) were obtained from Bioculture, Ltd. (Mauritius). MCM with at least one copy of the M1 MHC haplotype were chosen for this study. The macaques (n = 4) infected with SIV alone were cared for by the staff at the

Wisconsin National Primate Research Center (WNPRC) in accordance with the regulations, guidelines, and recommendations outlined in the Animal Welfare Act, the Guide for the Care and Use of Laboratory Animals, and the Weatherall Report. The University of Wisconsin-Madison (UW-Madison), College of Letters and Science and Vice Chancellor for Research and Graduate Education Centers Institutional Animal Care and Use Committee (IACUC) approved the nonhuman primate research covered under IACUC protocol G005507. The University of Wisconsin-Madison Institutional Biosafety Committee approved this work under protocol B00000205. All macaques were housed in standard stainless-steel primate enclosures providing required floor space and fed using a nutritional plan based on recommendations published by the National Research Council. Macaques had visual and auditory contact with each other in the same room. Housing rooms were maintained at 65–75°F, 30–70% humidity, and on a 12:12 light-dark cycle (ON: 0600, OFF: 1800). Animals were fed twice daily a fixed formula, extruded dry diet with adequate carbohydrate, energy, fat, fiber, mineral, protein, and vitamin content (Harlan Teklad #2050, 20% protein Primate Diet, Madison, WI) supplemented with fruits, vegetables, and other edible objects (e.g., nuts, cereals, seed mixtures, yogurt, peanut butter, popcorn, marshmallows, etc.) to provide variety to the diet and to inspire species-specific behaviors such as foraging. To further promote psychological well-being, animals were provided with food enrichment, structural enrichment, and/or manipulanda. Environmental enrichment objects were selected to minimize chances of pathogen transmission from one animal to another and from animals to care staff. While on study, all animals were evaluated by trained animal care staff at least twice daily for signs of pain, distress, and illness by observing appetite, stool quality, activity level, physical condition. Animals exhibiting abnormal presentation for any of these clinical parameters were provided appropriate care by attending veterinarians. Prior to all minor/brief experimental procedures, macaques were sedated with an intramuscular dose of ketamine (10 mg kg$^{-1}$) and monitored regularly until fully recovered from sedation. Per WNPRC standard operating procedure (SOP), all animals received environmental enhancement including constant visual, auditory, and olfactory contact with conspecifics, the provision of feeding devices which inspire foraging behavior, the provision and rotation of novel manipulanda (e.g., Kong toys, nylabones, etc.), and enclosure furniture (i.e., perches, shelves). At the end of the study, euthanasia was performed following WNPRC SOP as determined by the attending veterinarian and consistent with the recommendations of the Panel on Euthanasia of the American Veterinary Medical Association. Following sedation with ketamine (at least 15mg/kg body weight, IM), animals were administered at least 50 mg/kg IV or intracardiac sodium pentobarbital, or equivalent, as determined by a veterinarian. Death was defined by stoppage of the heart, as determined by a qualified and experienced individual.

MCM at the University of Pittsburgh (U.Pitt., n = 19) were housed in a BSL2+ animal facility during SIV infection and then moved into a BSL3+ facility within the Regional Biocontainment Laboratory for infection with Mtb. Animal protocols and procedures were approved by the U.Pitt. Institutional Animal Care and Use Committee (IACUC) which adheres to guidelines established in the Animal Welfare Act and the Guide for the Care and Use of Laboratory Animals, and the Weatherall report (8th Edition). The U.Pitt. IACUC reviewed and approved the IACUC study protocols 18032418 and 15035401, under Assurance Number A3187-01. The IACUC adheres to national guidelines established in the Animal Welfare Act (7 U.S.C. Sections 2131–2159) and the Guide for the Care and Use of Laboratory Animals (8th Edition) as mandated by the U.S. Public Health Service Policy. Macaques were housed at U.Pitt. in rooms with autonomously controlled temperature, humidity, and lighting. Animals were pair-housed in caging measuring 4.3 square feet per animal and spaced to allow visual and tactile contact with neighboring conspecifics. The macaques were fed twice daily with biscuits

formulated for nonhuman primates, supplemented at least 4 days/week with fresh fruits, vegetables or other foraging mix. Animals had access to water *ad libitem*. An enhanced enrichment plan is designed and overseen by our nonhuman primate enrichment specialist. This plan has three components. First, species-specific behaviors are encouraged. All animals have access to toys and other manipulanda, some of which are filled with food treats (e.g. frozen fruit, peanut butter, etc.). These are rotated on a regular basis. Puzzle feeders, foraging boards, and cardboard tubes containing small food items also are placed in the cage to stimulate foraging behaviors. Adjustable mirrors accessible to the animals stimulate interaction between cages. Second, routine interaction between humans and macaques are encouraged. These interactions occur daily and consist mainly of small food objects offered as enrichment and adhere to established safety protocols. Animal caretakers are encouraged to interact with the animals by talking or with facial expressions while performing tasks in the housing area. Routine procedures (e.g. feeding, cage cleaning, etc.) are done on a strict schedule to allow the animals to acclimate to a routine daily schedule. Third, all macaques are provided with a variety of visual and auditory stimulation. Housing areas contain either radios or TV/video equipment that play cartoons or other formats designed for children for at least 3 hours each day. The videos and radios are rotated between animal rooms so that the same enrichment is not played repetitively for the same group of animals. All animals are checked at least twice daily to assess appetite, attitude, activity level, hydration status, etc. Following SIV and/or Mtb infection, the animals are monitored closely for evidence of disease (e.g., anorexia, lethargy, tachypnea, dyspnea, coughing). Physical exams, including weights, are performed on a regular basis. Animals are sedated prior to all veterinary procedures (e.g. blood draws, etc.) using ketamine or other approved drugs. Regular PET/CT imaging is conducted on our macaques following Mtb infection and has proved very useful for monitoring disease progression. Our veterinary technicians monitor animals especially closely for any signs of pain or distress. If any are noted, appropriate supportive care (e.g. dietary supplementation, rehydration) and medications (e.g. analgesics) are given. Any animal considered to have advanced disease or intractable pain or distress from any cause is sedated with ketamine and then humanely euthanized using sodium pentobarbital (65 mg/kg, IV). Death is confirmed by lack of both heartbeat and pupillary responses by a trained veterinary professional.

## SIV and Mtb infection of MCM

At UW-Madison, 4 MCMs were infected intrarectally with 3,000 $TCID_{50}$ SIVmac239. Six months after infection, animals were humanely euthanized as indicated in the section above.

For studies performed at U.Pitt., animals in the SIV/Mtb co-infection group (n = 8) were infected intrarectally with 3,000 $TCID_{50}$ SIVmac239. After 6 months, the animals were co-infected with a low dose (3–12 CFU) of Mtb (Erdman strain) via bronchoscopic instillation, as described previously [33]. Animals in the SIV-naïve control group (n = 11) were infected with Mtb in an identical manner. TB progression was monitored by clinical testing and PT/CT imaging [33]. Six weeks after Mtb infection, animals were humanely euthanized, and necropsies were performed using PET/CT images to guide the excision of all granulomas. Random samples from each lung lobe were also harvested, as well as peripheral (axillary and inguinal) LNs, all thoracic LNs, mesenteric LNs, liver, and spleen [72].

## Sample collection

For all cohorts, whole blood and plasma samples were collected longitudinally pre- and post-SIV and/or Mtb infection, and PBMC and were isolated by Ficoll density gradient purification

as previously described [33, 73]. BAL samples were also collected at the indicated time points pre- and post-SIV and/or Mtb infection as previously described [33].

Tissue samples were collected at necropsy, as detailed above. For the SIV-only cohort, peripheral and thoracic LN were collected, single cell suspensions were prepared, and samples were stained for flow cytometry. For the SIV-naïve and SIV+ MCM infected with Mtb at U. Pitt, the tissue samples listed above were homogenized using Medimachines (BD Biosciences) and each homogenate was divided. One portion was plated on 7H11 agar to quantify the Mtb colony forming units (CFU) as previously described [33, 72], and the other portion was used for flow cytometry.

## Plasma viral loads

Plasma viral loads were quantified as previously described [73–76]. Briefly, viral RNA was isolated from plasma, reverse transcribed, and amplified with the SuperScript III Platinum one-step quantitative RT-PCR system (Thermo Fisher Scientific). Samples were then run on a LightCycler480 (Roche) and compared to an internal standard curve on each run.

## Flow cytometry

To characterize MAIT cells from peripheral blood, BAL, LN, and granulomas, the rhesus macaque (Mamu) MR1 tetramer loaded with either the 5-OP-RU active metabolite or the Ac-6-FP control metabolite were used [77]. The MR1 tetramer technology was developed jointly by Dr. James McCluskey, Dr. Jamie Rossjohn, and Dr. David Fairlie, and the material was distributed by the NIH Tetramer Core Facility as permitted to be distributed by the University of Melbourne. For PBMC, staining was performed using cryopreserved cells collected longitudinally from the current study, or cryopreserved cells from SIV-naïve MCM collected from previous, unrelated studies. For BAL and tissue samples, freshly isolated cell homogenates were used for staining.

Approximately $1x10^6$ cells (or all cells obtained from granulomas with $<1x10^6$ cells) were stained with 0.25 ug of Mamu MR1 5-OP-RU or Ac-6-FP tetramer for one hour in the presence of 500 nM Dasatinib (Thermo Fisher Scientific; Cat No. NC0897653). When TCRVα7.2 co-staining was performed, the antibody was added 30 minutes after the addition of the MR1 tetramer. Cells were washed once with FACS buffer (10% Fetal Bovine Serum (FBS) in a 1X PBS solution) supplemented with 500 nM Dasatinib, then surface antibody staining was performed for 20 minutes in FACS buffer + 500 nM Dasatinib. A complete list of antibodies used for surface staining is shown in Table 1. Samples were fixed in 1% paraformaldehyde for a minimum of 20 minutes. For intracellular staining, cells were washed twice with FACS buffer and staining with antibodies was performed in Medium B permeabilization buffer (Thermo Fisher Scientific, Cat. No. GAS002S-100) for 20 minutes at room temperature. For intranuclear staining, the True Nuclear Transcription Factor Buffer Set (Biolegend; San Diego, CA) was used according to manufacturer's instructions. Briefly, cells were fixed in the TrueNuclear fixation solution for 1 hour, then washed three times with the permeabilization buffer. Cells were then stained with the transcription factors indicated in Table 1 at 4°C for one hour, rinsed three times with permeabilization buffer, then resuspended in FACS buffer. Flow cytometry was performed on a BD LSR II (Becton Dickinson; Franklin Lakes, NJ), and the data were analyzed using FlowJo software for Macintosh (version 9.9.3 or version 10.1).

## Ex vivo analysis of MAIT function with partially fixed bacteria

MAIT cell function was measured *in vitro* by adapting assays that were previously described [52, 78]. Briefly, cryopreserved PBMC were rested for 6 hours in RPMI (Thermo Fisher

Scientific) supplemented with 10% FBS, 4 mM L-glutamine, and 1X antibiotic/antimycotic (Thermo Fisher Scientific Cat No. 15240062); this media will now be referred to as R10. For the stimulus, *E. coli* and *M. smegmatis* stocks were fixed for approximately three minutes with a 1% paraformaldehyde solution, washed three times with 1X PBS, and diluted to a working concentration of $1x10^6$ CFU/mL in R10. After the 6-hour resting period, approximately 10 CFU of the indicated fixed bacterium were added for each PBMC cell for 90 minutes to allow for bacterial uptake into the cells. After this 90 minute period, cells were then incubated for 16 hours at 37°C in the presence of 2 uM monensin (Biolegend; San Diego, CA), 5 μg/mL Brefeldin A (Biolegend; San Diego, CA), and CD107a-APC (Table 1). Cells isolated from LNs were incubated with bacteria for only 5 hours as the MAIT cell TCR is downregulated if the incubation period is too long. After incubation, cells were stained with the Mamu MR1-5OPRU tetramer in R10+500 nM Dasatinib (Thermo Fisher Scientific) for 1 hour. Cells were washed once with 1X PBS (Thermo Fisher Scientific) and live cells stained using the LIVE/DEAD NearIR staining kit (Thermo Fisher Scientific) at a 1:4,000 dilution in 1X PBS. Cells were washed once with PBS containing 10% FBS and then stained with CD3, CD4, and CD8 antibodies (Table 1). Cells were fixed in 2% paraformaldehyde for 20 minutes, permeabilized in Medium B, and stained with IFNγ and TNFα (Table 1). Flow cytometry was performed as indicated above.

## Statistical analysis

Pearson's correlation coefficients were calculated to determine the linear relationship between the absolute MAIT cell counts and the Log10 CFU within each granuloma. For comparative analysis of MAIT cell frequencies and phenotypes across cohorts, Mann-Whitney statistical tests were performed where indicated to determine statistical significance between two treatment groups, and Kruskall-Wallis tests with Dunn's adjusted p values were used to define statistical significance between three groups. For characterization of longitudinal changes across the same animals in a given cohort, repeated measures non-parametric ANOVA tests with Tukey's multiple test correction were performed to determine statistical significance. In situations where not all animals had longitudinally matched data (Figs 4B, 4C, 5D and Fig 8), Wilcoxon Rank signed tests were performed for matched pairs. To characterize longitudinal changes in the phenotypic markers present on MAIT cells during SIV infection, the data were first normalized to the average frequency for each marker before SIV infection for each animal. Then, the fold-change in expression of the indicated phenotypic marker relative to pre-infection measurement was calculated for every time point. Statistical significance was determined using non-parametric ANOVA tests with Tukey's multiple test correction. For granuloma/LN immunological data, means were calculated per animal and graphed to ensure that granulomas from individual animals were not biasing results. These graphs indicated that treatment effect sizes (of means) were similar to those in graphs where all granulomas/LN samples were pooled.

## Supporting information

**S1 Fig. Peripheral blood mononuclear cells (PBMC) and bronchoalveolar lavage (BAL) were collected from the designated timepoints post-SIVmac239 infection and flow cytometric analysis was performed as indicated in Figs 2 and 5.** Then, CD8+MR1tet+ cells were characterized for their expression of HLA-DR, TIGIT, ki67, CD39, and PD-1. Teal dots; CD8+MR1tet+ cells from BAL, Crimson dots; CD8+MR1tet+ cells from PBMC.
(EPS)

**S2 Fig. Representative gating schematic for MAIT cell functional assays.** Functional assays were performed as described in Methods and Figs 7 and 8, and flow cytometric analysis was performed. Top panels: representative gating schematic for CD8+MR1tet+ cells. Bottom 3 panels: CD107a (left), IFNγ, and TNFα were measured on CD8+MR1tet+ cells that were left unstimulated, or stimulated with 10 CFU/cell of either fixed *M.smegmatis* (*M.smeg*) or fixed *E. coli*. Total number of CD107a+, IFNγ+, and TNFα+ events within the CD8+MR1tet+ parent gate (total events) are indicated in the boxes to the left of the flow plots.
(EPS)

**S3 Fig. SIV and Mtb co-infection results in functional impairment of TNFα production in conventional CD8+ T cells in the blood.** A, Functional assays were performed as described in Fig 8 using PBMC from pre-Mtb (dark grey) and 6 weeks post-Mtb (yellow) infection from seven Mtb-infected MCM. Wilcoxon Rank sum statistical tests for matched pairs were performed to compare the expression of the indicated cytokines or CD107a longitudinally from conventional CD8+ T cells (CD8+MR1tet-) for a single stimulus. B, Functional assays were performed as described in (A) using PBMC from pre-SIV (dark grey), 20 weeks post-SIV (teal), and 6 weeks post-Mtb (yellow) infection from 8 SIV/Mtb coinfected MCM. Wilcoxon Rank sum statistical tests for matched pairs were performed to compare the expression of the indicated cytokines and CD107a longitudinally for a single stimulus. C, The functional assays performed in (A) and (B) using PBMC from six weeks post-Mtb infection for SIV-naïve (blue dots) and SIV+ (red dots) MCM were plotted on the same graphs to compare the expression of CD107a (first panel), IFNγ (second panel), or TNFα (third panel) amongst the two cohorts. Mann-Whitney tests were performed to determine statistical significance.
(EPS)

## Acknowledgments

We thank the animal care staffs at the University of Pittsburgh and the Wisconsin National Primate Research Center (WNPRC) for excellent care of the animals housed at these facilities. The content is solely the responsibility of the authors and does not necessarily represent the official views of the National Institutes of Health. We would like to thank the NIH tetramer core facility for providing the rhesus MR1-5OP-RU and 6-FP tetramers used in this study. Finally, we would also like to thank Virology Services, a member of Research Services, at the WNPRC for performing SIV viral load assays.

## Author Contributions

**Conceptualization:** Charles A. Scanga, Shelby L. O'Connor.

**Data curation:** Amy L. Ellis, Erica C. Larson, Pauline Maiello, Charles A. Scanga, Shelby L. O'Connor.

**Formal analysis:** Amy L. Ellis, Erica C. Larson, Pauline Maiello, Charles A. Scanga, Shelby L. O'Connor.

**Funding acquisition:** Charles A. Scanga, Shelby L. O'Connor.

**Investigation:** Charles A. Scanga, Shelby L. O'Connor.

**Methodology:** Amy L. Ellis, Alexis J. Balgeman, Erica C. Larson, Mark A. Rodgers, Cassaundra Ameel, Tonilynn Baranowski, Nadean Kannal, Charles A. Scanga, Shelby L. O'Connor.

**Project administration:** Charles A. Scanga, Shelby L. O'Connor.

**Resources:** Charles A. Scanga, Shelby L. O'Connor.

**Software:** Charles A. Scanga, Shelby L. O'Connor.

**Supervision:** Charles A. Scanga, Shelby L. O'Connor.

**Validation:** Amy L. Ellis, Erica C. Larson, Jennifer A. Juno, Charles A. Scanga, Shelby L. O'Connor.

**Visualization:** Amy L. Ellis, Erica C. Larson, Jennifer A. Juno, Charles A. Scanga, Shelby L. O'Connor.

**Writing – original draft:** Amy L. Ellis.

**Writing – review & editing:** Amy L. Ellis, Erica C. Larson, Pauline Maiello, Jennifer A. Juno, Charles A. Scanga, Shelby L. O'Connor.

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
