## [Decision Letter · Decision Letter 0]

3 Apr 2020

Dear Dr. O'Connor,

Thank you very much for submitting your manuscript "MAIT cells are functionally impaired in a Mauritian cynomolgus macaque model of SIV and Mtb co-infection" for consideration at PLOS Pathogens. As with all papers reviewed by the journal, your manuscript was reviewed by members of the editorial board and by several independent reviewers. The reviewers appreciated the attention to an important topic. Based on the reviews, we are likely to accept this manuscript for publication, providing that you modify the manuscript according to the review recommendations. The revised manuscript should incorporate all the modifications requested by the reviewers. No additional experiments are required. 

Sincerely,

Padmini Salgame

Associate Editor

PLOS Pathogens

Sabine Ehrt

Section Editor

PLOS Pathogens

Kasturi Haldar

Editor-in-Chief

PLOS Pathogens

orcid.org/0000-0001-5065-158X

Michael Malim

Editor-in-Chief

PLOS Pathogens

orcid.org/0000-0002-7699-2064

Reviewer Comments (if any, and for reference):

Reviewer's Responses to Questions

**Part I - Summary**

Reviewer #1: This manuscript entitled " MAIT cells are functionally impaired in a Mauritian cynomolgus macaque model of SIV and Mtb co-infection " characterizes circulating and lung MAIT cell phenotype and functions in SIV infected and mycobacterium tuberculosis infected macaques in an attempt to understand the impact of SIV infection on MAIT cell functions that could be protective against TB disease. The authors have rigorously conducted the detailed analyses of MAIT cell phenotype and functions in naïve and SIV infected macaques in order to address the impact during co-infection. The authors report that Mtb co-infection in SIV infected macaques contributed to a more activated or exhausted T cell phenotype with elevated levels of PD-1 and TIGIT expression than infection with SIV or Mtb alone. Further, following Mtb infection the MAIT cells from SIV-infected macaques are shown to be functionally impaired in their ability to produce TNFα when compared to SIV-naïve macaques. Overall, the manuscript is very well written and there is a good discussion of the main results, their implications in TB, and most importantly, the caveats of this study.

Reviewer #2: The investigators report the longitudinal effects of SIV (Wisconsin facility), Mtb mono-infection (Pittsburgh facility) and SIV/Mtb co-infection (Pittsburg facility) on macaque MAIT cell abundance and function. Overall, they report low numbers of MAIT cells in peripheral blood, BAL, lymph nodes in all experimental groups utilizing MR1-5OPRU tetramers and analyzing data using both % of T cells and absolute numbers. They also demonstrate that nearly all macaque MAIT cells are CD8+ and can be further subdivided into effector and central memory subsets based on canonical T cell surface receptors CD95 and CD28. They subsequently analyzed these subsets longitudinally ex vivo in peripheral blood (all infection models) and in BAL (SIV mono-infection) and found that MAIT cells transiently upregulate markers of activation/proliferation in the first 4-6 weeks after infection to some degree in all infection models. At the time of necropsy, MAIT cell numbers were found to be increased in the mediastinal lymph node of SIV mono-infected and SIV/Mtb co-infected macaques relative to granulomas. Lymph node MAIT cell number inversely correlated with Mtb CFU. Further, the authors show that mediastinal lymph node MAIT cells express increased activation/exhaustion markers PD1 and TGIT and decreased TNF�, a cytokine associated with control of Mtb infection.

Through a series of in vitro experiments, they demonstrate that healthy or infected macaque peripheral blood MAIT cells respond more robustly to E. coli lysates than those of M. smegmatis. Finally, they demonstrate that MAIT cell TNF� production is impaired after either E. coli or M. smegmatis re-stimulation in vitro in SIV/Mtb co-infected macaques.

Strengths:

This is the first study to compare MAIT cell abundance and function during SIV/Mtb co-infection and mono-infection in macaques. The authors identify distinct MAIT cell subsets based on effector/central memory markers and compare markers of activation/proliferation, chemokine receptors, and transcription factors with non-MAIT CD8+ T cells. They also demonstrate both ex vivo and in vitro functional differences between effector and central memory MAIT cells whose activity is transient during the first weeks after infection to some degree in all experimental models and wanes with no significant proliferation during the adaptive immune response. The authors demonstrate differential MAIT cell activity in vitro to re-stimulation with ligands from E. coli and M. smegmatis and report impaired MAIT cell function in SIV/Mtb co-infected animals.

This manuscript would be of great interest to the general MAIT cell and HIV/TB research community as it rigorously documents longitudinal changes to MAIT cell abundance and function during mono- and co-infection models of SIV and Mtb. The main findings of relatively low numbers of MAIT cells in all groups with transient expression of activation/proliferation markers early after infectious challenge is consistent with previous findings in Mtb mono-infected macaques (Bucson et al. Tuberculosis 2019) demonstrating early, but transient activation/proliferation. This data adds further evidence to support a limited role for MAIT cells in durable immunity against Mtb.

Reviewer #3: This is a well executed study of how MAIT cells in Mauritian cyno macaques are affected by SIV and Mtb infection. With one or two minor exceptions, the data are solid and well presented, and are valuable in their own right.

I have one significant quibble about an issue associated with the major hypothesis that drives the paper, include in many places but best stated on lines 268-270 at the beginning of the introduction: "Here, we wanted to test the hypothesis that SIV-dependent dysregulation of MAIT cells weakened their ability to respond to a mycobacterial infection, thus failing to contain Mtb and leading to a rapid early increase in the number of granulomas in co-infected animals." While I think this is a reasonable hypothesis, and despite that all the experiments are about MAIT cells, I'm not sure any of the experiments really address the hypothesis (and in fact, doing so in NHPs, with no knockouts and little ability to selectively deplete MAITS, is very difficult). I don't think this undermines the value of the data, which is still strong, but the hypothesis remains untested.

**Part II – Major Issues: Key Experiments Required for Acceptance**

Reviewer #1: (No Response)

Reviewer #2: Inherent in the macaque model is the small sample size, which inevitably highlights questions of interindividual variability and reproducibility. The SIV mono-infection model and Mtb mono-infection/SIV/Mtb co-infection models were conducted at different facilities with different breeding practices/environmental microbiota which are variables that remain uncontrolled. The authors demonstrate that nearly all macaque MAIT cells are CD8+, but in light of literature identifying both CD4+ and CD4-CD8- MAIT cell subsets in human primates, the manuscript would be strengthened if clearly stated that non-CD8+ MAIT cells were not identified in any macaques. This should also be discussed in the setting of murine animal models where most MAIT cells lack CD4 or CD8 co-expression.

The authors nicely compare immune phenotypes of central/effector memory MAITs with non-MAIT CD8+ T in the first figure. However, this comparison approach was abandoned in later figures where only MAIT cell functional data is reported (eg, fewer TNF�+MAIT detected in SIV/Mtb co-infected mediastinal lymph node MAIT cells). It would greatly strengthen the manuscript if the same functional analyses were performed in non-MAIT CD8+ T cells in order to assess whether this immune dysfunction is unique to the MAIT cell subset or potentially common to conventional CD8+ T cells.

Finally, the authors demonstrate that SIV and SIV/Mtb infected macaques have increased numbers of MAIT in mediastinal lymph nodes that inversely correlate with Mtb CFU and that demonstrate upregulated markers of exhaustion and decreased TNF� production. The manuscript would be greatly strengthened if chemokine receptor expression (eg, Figure 1) was characterized on these cells, as this could suggest a mechanism by which SIV may inhibit trafficking to pulmonary sites of infection leading to increased mortality.

Reviewer #3: No experiments are required.

**Part III – Minor Issues: Editorial and Data Presentation Modifications**

Reviewer #1: Specific comments:

1. The concluding lines in the abstract state that, “the impact of SIV-dependent dysregulation of MAIT cell function on TB disease course was unclear because these animals were euthanized just 6 weeks after Mtb infection. However, their dysregulation by SIV could disrupt control of Mtb replication”. It would be better to focus it on the power of this study, which is that tissue/granuloma specific responses were studied early on during co-infection, rather than stating what was not accomplished due to the study design…It may be nicer to rephrase it focusing on this aspect, for example:

Thus, SIV-mediated dysregulation of MAIT cell functions occurs early following Mtb-coinfection, and has the potential to disrupt control of Mtb replication in the lungs over the course of SIV/TB co-infection.

2. Line 73 in Introduction section, the rationale for using Mauritian cynomolgus macaques (MCM) is given as having simplified major histocompatibility complex (MHC) genetics. The authors need to clarify on why that is an advantage for this study to make it easier for readers from outside the nonhuman primate (nhp) field to understand the rationale, especially since most other nhp studies have used Indian rhesus macaques or the Chinese cynomolgus macaques.

3. There is a lack of consistency in the gating for MAIT cells between figures 1, 3 and 5. Fig.1 B shows distribution of TCR Va7.2 and CD4 and CD8 on MR1 tet+ cells and the tet- T cells. Please label the y-axis as %TCR Va7.2 instead of TCR only. Also, it is not clear that the % CD4 or CD8 expression inside the box on the right is from total CD3+ tet+ cells, or MR1tet+TCRVα7.2+ cells, although the legend mentions the latter. It would be simpler to just add a label as MR1tet+TCRVα7.2+ cells on the box that shows CD4, CD8 expression and memory phenotype. Based on Fig 1, the MAIT cells are from CD3+ MR1 5OPRU tet+ cells (Fig 1 A, B). Then in Fig 3, 4 and 6, the MAIT cells are gated on CD8+ cells instead of CD3+ MR1 5OPRU tet+ cells shown in the top panel of Fig 3. Although all of the tet+ cells are in the CD8hi population, the denominator (total CD3 cells or CD8+ T cells) can impact the cell counts, especially in SIV infection when CD4/CD8 ratios change. In Fig 5, we are back to CD3+ MR1 5OPRU tet+ cells. It would be good to stick to one gating strategy to define MAIT cells throughout the study.

Reviewer #2: (No Response)

Reviewer #3: I'll give these in two bullet lists, one about figures, and one about text.

Notes on Figures:

* Figure 1A - on the right most panel, please plot MAIT frequency vs age

* Figure 1B - on right most plot, replace bar graph with dots and quartiles

* Figure 3B, C, D - add color coding to symbols

* Figure 4A highlights LN and granuloma collection, but the data in the rest of the figure appear to be about PBMC. This is just a bit strange organizationally. It looks like all the LN and granuloma stuff will be analyzed in a future manuscript.

* Figure 5. Add comparisons to blood phenotypes if data are available. Explore a looser gate on FSC-SSC to catch potential blasts.

* Figure 7: I’d like to see sample a flow cytometry plots of MR1tet+ vs each of CD107, IFNg, and TNFα gated on CD8+ cells corresponding to data in panel A.

* Figure 7: are there complications when comparing E. Coli vs M.smeg stimulations, including potential differentlal stimulation of innate cytokine responses that have nothing to do with MR1-restricted recognition. In addition, the PBMC used were frozen — this probably makes a difference with monocyte APC function. There might also be a complication with MR1 expression and MAITs responding to cytokines rather than riboflavin precursors. Also, protocol as written is flawed because monensin is added at the same time as bacterial -- this is likely to limit transit of MR1 to the cell surface.

Notes on Text

* Line 20: The statement here (and similar ones elsewhere) makes an assumption not proven in the manuscript, that MAIT cells contribute to control of Mtb. (Also discussed above)

* Line 51: substitute “invariant” for “innate” (which is a debatable term here).

* The paragraph beginning on Line 61 doesn’t indicate why a population specific for riboflavin precursors should be affected by HIV or SIV infection.

* Lines 74-77. The connection between granuloma numbers and impaired immune responses is not obvious.

* Line 134-137. Were increases in markers such as Ki67 due to cytokine receptors on MAITs or TCR? Probably hard to determine in MCM, but this fact should be noted.

* Line 145-146. Specify Mtb strain within the text here.* Lines 236-236 could use a reference for the pHRODO assay.

* Line 290. The relationship between MAIT derived TNFα production and Mtb control is not established by this paper. Could be other T cells. This whole paragraph should be toned down a bit. This is acknowledged on line 311.

PLOS authors have the option to publish the peer review history of their article (what does this mean?). If published, this will include your full peer review and any attached files.

Reviewer #1: Yes: Namita Rout

Reviewer #2: No

Reviewer #3: No
---

## [Editor Report · Decision Letter 1]

29 Apr 2020

Dear Dr. O'Connor,

We are pleased to inform you that your manuscript 'MAIT cells are functionally impaired in a Mauritian cynomolgus macaque model of SIV and Mtb co-infection' has been provisionally accepted for publication in PLOS Pathogens.

Best regards,

Padmini Salgame

Associate Editor

PLOS Pathogens

Sabine Ehrt

Section Editor

PLOS Pathogens

Kasturi Haldar

Editor-in-Chief

PLOS Pathogens

orcid.org/0000-0001-5065-158X

Michael Malim

Editor-in-Chief

PLOS Pathogens

orcid.org/0000-0002-7699-2064
---

## [Editor Report · Acceptance letter]

13 May 2020

Dear Dr. O'Connor,

We are delighted to inform you that your manuscript, "MAIT cells are functionally impaired in a Mauritian cynomolgus macaque model of SIV and Mtb co-infection," has been formally accepted for publication in PLOS Pathogens.

Best regards,

Kasturi Haldar

Editor-in-Chief

PLOS Pathogens

orcid.org/0000-0001-5065-158X

Michael Malim

Editor-in-Chief

PLOS Pathogens

orcid.org/0000-0002-7699-2064